# Intraoperative Tension Pneumothorax in a Trauma Patient: An Adult Simulation Case for Anesthesia Residents

**DOI:** 10.3390/healthcare10091787

**Published:** 2022-09-16

**Authors:** David Ryusuke Okano, Andy W. Chen, Sally A. Mitchell, Johnny F. Cartwright, Christopher Moore, Tanna J. Boyer

**Affiliations:** 1Department of Anesthesia, Indiana University School of Medicine, Indianapolis, IN 46202, USA; 2Indiana University School of Medicine, Indianapolis, IN 46202, USA

**Keywords:** simulation, anesthesiology, pneumothorax, crisis resource management, communication

## Abstract

Anesthesiologists may encounter multiple obstacles in communication when attempting to collect information for emergency surgeries. Occult tension pneumothorax that was asymptomatic in the emergency department (ED) could become apparent upon positive pressure ventilation and pose a critical threat to the patient intraoperatively. Here, we describe a simulation exercise that was developed as a curriculum module for the Indiana University (IU) Anesthesiology residency program. It is primarily designed for first-year clinical anesthesia residents (CA-1/PGY-2). It is a 50 min encounter with two scenarios. The first scenario focuses on information collection and communication with a non-cooperative patient with multiple distractors. The second scenario focuses on the early diagnosis of tension pneumothorax and subsequent treatment. The residents were given formative feedback and met the educational objectives. Commonly missed critical actions included misdiagnosing the tension pneumothorax as mainstem intubation, bronchospasm, pulmonary thromboembolism, and anaphylaxis. Residents rated the feedback and debriefing as “extremely useful” or “very useful.” Time constraints limit the number of residents who can sit in the “hot seat.” The structure of the mannequin limits the ability to diagnose pneumothorax by auscultation and ultrasound. In the future, the scenarios may also be utilized to educate student anesthesiologist assistants and other non-physician anesthesia learners.

## 1. Introduction

Trauma patients who require emergency surgery are often not optimized for anesthesia. There will be many challenges for the anesthesiologist in the emergency department (ED), such as surgeons wanting to rush the patient to the operating room (OR). Additionally, the patient may be agitated, intoxicated, or unconscious. A detailed medical history is not always available. Information from the emergency medical technician (EMT) could be lost or misunderstood during multiple hand-offs. Focused exams and lab work are often incomplete or unable to be obtained. Minor issues could be missed due to issues that require immediate attention. Anesthesiologists must collect as much information as possible to evaluate and plan for the safest anesthesia plan in a short timeframe. Despite the best effort, unanticipated complications may present in the OR.

Tension pneumothorax is a life-threatening complication that has been suggested to occur in 5.4% of major trauma patients in the prehospital environment [1]. It is a type of pneumothorax with pleural damage acting as a check-valve, resulting in air trapping in the ipsilateral thoracic cavity that displaces the mediastinum and compresses the contralateral lung. This causes an acute onset of hypoxia and a critical compromise of the cardiovascular system. Patients with chest trauma with adequate ventilation and oxygenation may later develop tension pneumothorax intraoperatively under positive pressure ventilation [2]. Tension pneumothorax is known to have a high mortality rate; one retrospective cohort study in an ICU setting found mortality to be 38 times higher for tension pneumothorax patients on mechanical ventilation compared to those who were not [3]. Another more recent retrospective cohort study also in an ICU setting found the development of tension pneumothorax among mechanically ventilated patients to be associated with an increase in mortality by a hazard ratio of 7.4 [4]. Needle decompression provides immediate relief of the trapped air and achieves the restoration of oxygenation and circulation.

This simulation was developed to train anesthesiology residents on how to quickly collect relevant information in a distracting environment and establish the safest anesthetic plan for emergency surgery. Residents also learned how to recognize the clinical manifestation of tension pneumothorax, make the diagnosis, and properly treat this life-threatening complication in a timely manner while maintaining efficient communication with the perioperative team.

## 2. Materials and Methods

The case is fully presented for facilitators in the Simulation Case file (Appendix A). The Simulation Case file consists of Simulation Case Overview (Table A1), Initial Presentation (Table A2), Instructor Notes—Changes and Case Branch Points (Table A3), and Instructor Notes—Ideal Scenario Flow and Anticipated Management Mistakes (Table A4). A critical actions checklist (Appendix B) is included for learners to reference during the simulation. A debriefing handout (Appendix C) is included to facilitate the post-assessment session.

### 2.1. Educational Objectives

By the end of this activity, learners will be able to:Collect pertinent medical history from a trauma patient that is not optimized for the proposed surgery, where the patient–physician interaction occurs in the ED with a distracting environment.Communicate effectively with surgeons who are trying to rush the patient to the operating room (OR).Communicate with family members if the patient is considered not eligible to consent.Diagnose the intraoperative tension pneumothorax by identifying the signs and symptoms and discuss the differential diagnosis of tension pneumothorax.Perform a needle decompression to treat the critical cardiopulmonary decompensation caused by tension pneumothorax.

### 2.2. Equipment and Environment

This two-part simulation required two rooms: an ED patient holding area for the first scene, and an OR for the second scene. For the first scene, the patient holding area can be any room similar to an ED room with a patient bed and a simulated monitor. For the second scene, the OR can be either real (in situ) or a simulated laboratory. We used an in-hospital decommissioned space with a mock OR and patient room for the ED. The mannequin, operating table, monitors, and equipment were available in the room, while the programming and monitoring of the scenario were accomplished via wireless audio, video feed, and remote control.

For the ED scene, the patient actor wore a hospital gown and lay on the ED bed with bandages around their head and right thigh. The patient had one prop peripheral IV in the upper extremity and wore a neck collar and simple oxygen mask. An EKG monitor, blood pressure cuff, and pulse oximeter were hooked up to the patient, and the data were displayed on the simulated monitor. A cordless phone was available to call the patient’s family member (embedded participant) to obtain anesthesia consent. A prop food tray was prepared and set outside the ED room.

For the OR scene, the mannequin was positioned supine on the operating table with a neck collar and one peripheral IV in the upper extremity. To save time, the surgical drape was already in place over the mannequin with a fenestration at the right femoral region. We used the Laerdal SimMan 3G simulation model connected to IngMar ASL 5000 Lung Solution.

In the OR, an anesthesia machine for simulation was available, as well as a fully equipped anesthesia cart with a video laryngoscope, stethoscope, and standard set of prop anesthesia drugs. A chest drain and chest tube were made available to the surgeon actor. A code cart was available in the hallway just outside of the OR.

### 2.3. Personnel

Role assignments for the simulation can be very flexible depending on the number of participants. At a minimum, three participants/roles are required: one simulation learner as the anesthesia provider, one embedded participant as the patient actor in the trauma bay, and one embedded participant as the surgeon actor. In our program, we typically had 3 to 4 anesthesiology residents participate as a group. One resident was in the “hot seat” as the anesthesiologist and the others chose an embedded participant role. The number of participants could be increased to add realism and value to the simulation as a multidisciplinary educational activity. Extra participants could be assigned as resident surgeons, scrub nurses, circulator nurses, or a meal service aid. Simulation operation specialists programmed and ran the high-fidelity mannequin and simulated patient monitors. The instructor was behind the scenes during the simulation scenario to give cues and instructions through the headset worn by the surgeon.

### 2.4. Implementation

First, the patient actor was prebriefed on their role (Appendix D) and then waited in the ED holding area. The surgeon actor was prebriefed (Appendix E), donned the headset, and waited outside the holding area. The learner anesthesiologist was prebriefed (Appendix F) and instructed to become familiar with the simulated OR, including setting up the anesthesia equipment and drugs for the proposed emergency surgery. Next, the learner was instructed to evaluate and consent the patient in the ED holding area.

The trauma patient reported riding a motorcycle under the influence of illicit drugs with their “buddy” when involved in a traffic accident, suffering a compound fracture of the right femur with moderate damage to the femoral vein. Temporary hemostasis was achieved by EMS during transfer to the ED. As the learner anesthesiologist entered the room, the patient was awake and alert, complaining of some pain in the head, neck, chest, and the open wound of the right thigh, but is not in apparent distress. The patient is restless and appears to be more concerned about their “buddy” who was riding on the back seat of the motorcycle. The patient is a very poor historian and cannot give detailed past medical history but admits to habitual use of illicit drugs. A preoperative chest X-ray (Appendix G) was available upon request.

Multiple distractions occurred while the anesthesiologist attempted to evaluate the patient and obtain consent. The surgeon entered the room and began talking to the patient, interrupting the anesthesiologist. A “Code Blue” was announced overhead, and the patient became agitated suspecting it might be about their “buddy.” A food tray was delivered to the room by mistake and the hungry patient tried to reach for it. The circulator nurse called the anesthesiologist’s cell phone to ask how many units of blood should be ordered. Some of these distractions may be omitted depending on the learner anesthesiologist’s communication skill level, as well as time and/or personnel constraints.

If the learner anesthesiologist attempted to postpone the case because the patient was not able to consent, the surgeon would insist the surgery could not be postponed since the patient has a compound fracture with damage to the blood vessel. The surgeon would suggest the anesthesiologist call the family member to obtain phone consent.

After obtaining the phone consent, the anesthesiologist and the surgeon entered the OR. The patient actor donned the headset and switched their role to either resident surgeon or a circulator nurse, then entered the OR as well.

Once all participants were in the OR, the learner anesthesiologist was instructed to verbalize their train of thought so that the instructor could anticipate how the scenario may progress. Administration of medications was accomplished by verbalizing the action, but this could be accompanied by pushing the prop medications to the mannequin if it has the capability.

After the time-out protocol, general anesthesia was induced. The learner anesthesiologist should consider rapid sequence induction with in-line stabilization of the cervical spine given the patient’s condition. Video laryngoscope could be utilized depending on the learner anesthesiologist’s skill level.

Once the airway was secured and the drapes were up, the surgeon announced the start time and incision. As positive pressure ventilation continued, the patient’s vital signs gradually deteriorated. Oxygen saturation dropped to the low 70′s, heart rate (HR) increased up to the 130 s, and systolic blood pressure (SBP) dropped to the 60′s. End-tidal carbon dioxide (ETCO_2_) decreased to 28 mmHg, and peak airway pressure rose to 45 cmH_2_O. The anesthesiologist should recognize the deterioration and immediately start communicating with the surgical team. They analyzed the abnormal vital signs and collected more information to eventually make the diagnosis of tension pneumothorax. The anesthesiologist should also notice that there are no lung sounds heard on the right side upon auscultation.

Once the diagnosis of tension pneumothorax was made, the learner anesthesiologist performed a needle decompression. The anesthesiologist could mimic this action, or they may insert an IV catheter into a specified chest port if the mannequin (e.g., Laerdal SimMan 3G) has this functionality. If the learner anesthesiologist is unsure where to place the needle on the mannequin, the instructor could offer guidance through the surgeon’s headset using the surgeon actor as a proxy.

Once the needle decompression was accomplished, the vital signs improved. The surgeon offered to place a right chest tube and hook it up to the drain. An intraoperative chest X-ray (Appendix H) could be ordered to confirm the reinflation of the lung and the correct placement of the chest tube. The scenario ended when the anesthesiologist noticed the improvement of the patient’s clinical condition and determined the post-operative disposition. The anesthesiologist decided if the patient was to be extubated or left intubated and transferred directly to the ICU. There was no absolute right or wrong answer so long as the learner anesthesiologist provided an appropriate rationale and justification.

### 2.5. Assessment

The facilitator reviewed the completion of the Critical Actions Checklist (Appendix B). Learners received formative feedback during the debriefing.

### 2.6. Debriefing

The debriefing was held immediately following the conclusion of the scenario. All the participants, including observers, moved to a nearby classroom. Alternatively, they could have remained in the simulated OR for the debriefing. The debriefing session began with an open-ended question to the learner who was in the “hot seat” about how they felt the scenario went. This is a strong tool to elicit spontaneous reflections from the learner as well as stimulate active discussions from the perspective of other participants. The following points are recommended to be discussed during the debriefing with the learners (Appendix C):Evaluation of the patient in ED: This scenario involves multiple communication barriers during the assessment of the patient. The learner anesthesiologist should discuss how to stay focused in order to determine the best and the safest anesthesia plan in the middle of multiple distractors. Discuss if the anesthesiologist missed any information due to the distractions. Participants may share their real-life experiences in dealing with similar situations.Communication with the surgical team: The learners should discuss the feasibility of obtaining anesthesia consent from a patient with questionable mental status. Where do you draw the line? What should you do if the family member or surrogate decision-maker is not available for consent? Additionally, discuss how to communicate with the surgeon in an assertive manner if you have any concerns about proceeding with the case.Identification of tension pneumothorax: The learner anesthesiologist should discuss the differential diagnosis for the collective findings of desaturation, tachycardia, hypotension, and increased peak airway pressure. What are the differences between anaphylaxis, pulmonary thromboembolism, bronchospasm, and tension pneumothorax? Pros and cons of additional diagnostic measures such as chest auscultation, lung POCUS, and chest X-ray should be discussed.Treatment of tension pneumothorax: The learners should discuss the pathophysiology and treatment of tension pneumothorax. Both the recommended needle size and the insertion site location for the needle decompression should be emphasized.Postoperative airway management planning: The learners should discuss whether the tension pneumothorax patient who has already received definitive treatment with the chest tube could be extubated or would need to remain intubated after the surgery. It should be noted that many thoracic surgery patients can be safely extubated immediately after the procedure.

## 3. Results

The anesthesiology residency program at the Indiana University School of Medicine Department of Anesthesia has approximately 28 residents per class. We conduct simulation training for residents on a regular basis. Every Thursday, a group of 3 to 5 residents is assigned to a 3-h-long simulation training block. Each block consists of 3 to 4 anesthesia simulation scenarios that last 40 to 50 min each. The contents range from procedural workshops on perioperative anesthetic management to Objective Structured Clinical Exam (OSCE) training which puts more focus on interpersonal communication. This 50 min scenario was developed in 2013 and has been implemented yearly until the present year (2022) to train CA-1 anesthesia residents who are typically 8 to 9 weeks into their residency training. Two simulation faculty members have been involved in running this scenario, which has been carefully designed and updated annually to maintain an appropriate quality of education.

We have tried to keep our simulation experience as realistic as possible. Participants were asked to wear proper attire for each role. Supplies such as scrubs, masks, hats, gloves, patient gowns, and surgical gowns were available at the simulation center. Although the CA-1 residents are still in the very early stages of their training, they could relate very well to the distractors we have embedded in the scenario. They reported experiencing similar clinical situations as medical students and interns. Residents reported that they enjoyed playing the roles of the patient and the surgeon.

We usually do not grade our residents’ performance during the simulation training. We try to provide carefully selected perioperative incident topics relevant to the practice of anesthesia. Simulation faculty members ensure the learners are allowed to make mistakes without feeling psychological and emotional stress. Residents can learn from the mistakes committed during the simulations and be prepared to react to similar real-life incidents in the future.

Most of our residents met the majority of the educational objectives by performing very well in communicating with the patient, surgeon, and the family member of the patient. Most of them performed the induction sequence smoothly by paying adequate caution to a potentially full stomach and a difficult airway from wearing a neck collar. Some residents initially did not reach the diagnosis of tension pneumothorax and reported thinking it was either mainstem intubation, bronchospasm, pulmonary thromboembolism, or anaphylaxis. Instructors were able to redirect the learner anesthesiologist by giving suggestions through the surgeon actor.

Residents were asked to provide feedback after the completion of each simulation. A total of 25 residents completed the electronic post-simulation questionnaire, representing a response rate of 31%. Each item of the survey was rated on a 5-point Likert scale (e.g., 1 = strongly agree, 2 = somewhat agree, 3 = neither agree nor disagree, 4 = somewhat disagree, 5 = strongly disagree). The two lowest (e.g., 1 = strongly agree, 2 = somewhat agree) and two highest (e.g., 4 = somewhat disagree, 5 = strongly disagree) categories were combined for data reporting (Table 1). The mean and standard deviation values were calculated using the raw Likert scores described above.

Items 1 and 2 assessed the residents’ baseline confidence in their ability to identify intraoperative tension pneumothorax and perform needle decompression, respectively, prior to engaging in the simulation activity. Some learners did not feel confident in their ability to perform these tasks, as items 1 and 2 had “disagree” response rates of 16% and 48%, respectively. However, regardless of the learners’ baseline skills, most residents indicated that the simulation improved their confidence in performing these tasks, as indicated by the “agree” response rates of 92% and 88% for items 3 and 4, respectively. Items 5, 6, and 7 assessed the faculty debriefing session and were scored highly by the residents, with no negative scores recorded. All participants indicated that the feedback and debriefings were “extremely useful” or “very useful.”

Qualitative data were collected from open-response items. Participants were asked to offer any constructive criticism or suggestions for improvement. Two of 25 residents responded with free-text: (a) “Thought it was great. Sim is very useful,” and (b) “Good session with adequate debriefing. It was an engaging session and I found it beneficial.”

## 4. Discussion

Simulation education is vital in training anesthesia residents to manage perioperative crises—arguably the most important part of their jobs as future anesthesiologists. However, simulation in healthcare tends to focus only on the acquisition of cognitive and procedural skills. The main purpose of the development of this simulation was to help residents develop crisis management skills for potential perioperative incidents in a safe environment. A secondary purpose was to foster excellent communication skills with patients and the perioperative team as mature anesthesiologists.

We also emphasized communication skills because it augments realism and enhances learning. The surgeon-anesthesiologist relationship is frequently viewed as hierarchical, which lends to debriefing residents on advocacy and assertive strategies to invoke when a team member’s (anesthesiologist) viewpoint does not align with that of decisionmaker [5]. TeamSTEPPS^®^ recommends stating the concern, problem, and solution in a firm and respectful manner [5]. If the initial statements are ignored, the next recommendation use the Two-Challenge Rule—voice the concern a second time and ensure the person being challenged acknowledges hearing the concern [5]. If the issue remains yet unresolved, then making stronger assertive statements using CUS (I am concerned; I am uncomfortable; This is a safety issue; Stop the line!), utilizing the chain of command, or calling a supervisor/leader may necessary [5].

Due to time constraints, one of the limitations of this simulation is that only one or a pair of residents from a group of 3 to 5 can be placed in the anesthesiologist’s “hot seat” at most, and the rest of the residents from that group must participate as non-anesthesiologist roles. Although feedback from the residents playing non-anesthesiologist roles indicated that they also learned much from the simulation, it remains to be seen if there is a difference in the effectiveness of training between the residents who are in the anesthesiologist “hot seat” and those who are not. Of note, the post-survey showed most residents agreed that their confidence level and knowledge increased, and all residents agreed that the debriefing was useful. To follow up longitudinally, we plan to survey the residents as to their experiences and the frequency of encountering pneumothorax cases in a variety of clinical settings, including perioperative, OR, ICU, and emergency room. This may serve as validity evidence to support continued simulation sessions, inform realism improvements, and promote self-reflection.

Imaging studies such as a chest X-ray, CT, or point-of-care ultrasound (POCUS) can confirm the diagnosis of pneumothorax. However, given the life-threatening nature of tension pneumothorax, it has been recommended that needle decompression be performed emergently rather than waiting on diagnostic imaging studies [6,7], basing the diagnosis primarily on clinical manifestations such as hemodynamic compromise, hypoxia, absent unilateral breath sounds on auscultation, and mediastinal shift [1,6]. The IngMar ASL 5000 Lung Solution connected to the Laerdal SimMan 3G simulation model can mimic spontaneous respiratory air movements, but it cannot simulate the ventilation dynamics of pneumothorax. Although SimMan 3G is capable of reproducing the unilateral lung ventilation sound required for this scenario, the sound from the speaker in one lung reverberates within the thoracic cavity of the mannequin and can be heard on the contralateral side. This was another limitation of this simulation, as it caused some residents to fail to diagnose pneumothorax by auscultation.

To address these issues, we plan to improve the scenario by adding an ultrasound machine to the available equipment list and providing digital medical images/videos of an E-FAST exam (Extended—Focused Assessment with Sonography for Trauma) since the residents cannot perform the exam on the mannequin. The FAST provides four basic views (the four “Ps”: pericardial, perihepatic, perisplenic, and pelvic); the extended provides two views of the thorax to assess for pneumothorax and hemothorax [8,9,10]. As the use of POCUS increases in the perioperative environment, it is important to learn from other disciplines that pneumothoraces are common in trauma patients, and that an estimated 50% are missed on routine chest X-ray [11]. Thus, the importance of teaching anesthesiology residents to perform and interpret an E-FAST exam should be recognized by both simulationists and clinical educators.

Althoughother simulation exercises revolving around chest tube placement have been described [12,13], they weredesigned with the primary goal of developing this specific procedural competency within the learners—usually emergency medicine or surgery residents. Our simulation exercise was directed toward anesthesiology residents and placed greater emphasis on communication dexterity within the perioperative setting (with the distractors built into Scene 1) as well as the synthesis of data gathered by intraoperative monitoring methods to diagnose tension pneumothorax (with the OR in Scene 2).

Our anesthesia department also has an Anesthesiologist Assistant (AA) training program. We believe their training will be enhanced further when we are able to offer similar training to student AAs (SAAs) in the future. Since one of the themes of this simulation scenario is communication with patients and the perioperative team, it would make the simulation experience more realistic and effective if participants from different specialties, such as nursing and surgery, and standardized patient actors can join on a regular basis.

## Figures and Tables

**Table 1 healthcare-10-01787-t001:** Distribution of Survey Responses, Item Mean Scores, and Standard Deviations (N = 25).

	Percent % (Raw Count)				
Item ^1^	Agree ^2^	Neutral ^3^	Disagree ^4^	M ^5^	SD
Before this simulation session, I could confidently IDENTIFY intraoperative Tension Pneumothorax.	56% (14)	28% (7)	16% (4)	2.44	0.94
2. Before this simulation session, I could confidently perform NEEDLE DECOMPRESSION for intraoperative Tension Pneumothorax.	32% (8)	20% (5)	48% (12)	3.20	1.26
3. My confidence in how to IDENTIFY intraoperative Tension Pneumothorax has improved as a result of this simulation session.	92% (23)	8% (2)	0% (0)	1.44	0.64
4. My confidence in how to perform NEEDLE DECOMPRESSION for intraoperative Tension Pneumothorax has improved as a result of this simulation session.	88% (22)	12% (3)	0% (0)	1.44	0.70
5. The debriefing faculty created a psychologically safe learning environment throughout the debriefing session.	96% (24)	4% (1)	0% (0)	1.12	0.43
6. I had the opportunity to ask questions during the debriefing session.	100% (25)	0% (0)	0% (0)	1.08	0.27
7. I received useful feedback and the most important issues were summarized during the debriefing sessions.^6^	100% (25) ^7^	0% (0) ^8^	0% (0) ^9^	1.20	0.40

Abbreviations: M, Mean; SD, Standard Deviation. ^1^ Rated on a 5-point Likert scale (1 = *strongly agree*, 2 = *somewhat agree*, 3 = *neither agree nor disagree*, 4 = *somewhat disagree*, 5 = *strongly disagree*). ^2^ Strongly agree, somewhat agree. ^3^ Neither agree nor disagree. ^4^ Somewhat disagree, strongly disagree. ^5^ Mean values were calculated using raw Likert point scores. ^6^ Rated on a 5-point Likert scale (1 = *extremely useful*, 2 = *very useful*, 3 = *moderately useful*, 4 = *slightly useful*, 5 = *not at all useful*). ^7^ Extremely useful, very useful. ^8^ Moderately useful. ^9^ Slightly useful, not at all useful.

## Data Availability

The evaluation forms electronically filled in by the participating residents are stored on Indiana University Qualtrics website. The website is not open to the public. Please contact the corresponding author for access to survey data.

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
