# Peer review of "Intraoperative Tension Pneumothorax in a Trauma Patient: An Adult Simulation Case for Anesthesia Residents"

_healthcare, 2022, doi:10.3390/healthcare10091787_

Round 1
Reviewer 1 Report
MS No: Healthcare-1899991
MS Title: Intraoperative Tension Pneumothorax …………
MS Authors: David R. Okano et al.
20220829
This is a well-written article about experience of using a medical simulation training program to evaluate and enhance the novice anesthesia residents’ performance and competency specifically during clinical scenario of tension pneumothorax. Study design includes objectives (collection of clinical information, identification crisis, and communication skills) before and during anesthesia and surgery. Such simulation exercise was developed as a curriculum module for anesthesia residency program (CA-1/PGY-2). The study subjects were given formative feedback, and assessed the educational objectives (including any missed critical actions) and debriefing. Few limitations of this study were mentioned. All together, this article shows a well-defined medical simulation protocol for teaching anesthesia residents.
Comments:
1. Page 2, line 53: Cited reference-3 is about pneumothorax in ICU. Any citation about studies on (tension) pneumothorax intra-operatively?
2. Page 6: Table 1: Any quantitative analysis and statistical conclusion of the observed results (n=25)?
3. Page 7, line 292: Any further analysis of the qualitative data?
4. Page 7, line 311: Indeed, the validity of this simulation protocol remains to be determined. How will the authors validate this teaching model in the future? The authors do indicate two possible limitations in this study.
5. Are there any similar simulation protocol regarding pre-op and peri-op occurrence of tension pneumothorax for residents in literature? For example,
a. American Journal of Roentgenology. 2012;199: 244-251. 10.2214/AJR.11.7892;
b. https://doi.org/10.15766/mep_2374-8265.11266
c. Adv Simul 1, 21 (2016). https://doi.org/10.1186/s41077-016-0021-2
d. MEDICAL EDUCATION 2003;37(Suppl. 1):14–21
e. etc
6. In addition, since there are similar medical simulation programs for residents to learn tension pneumothorax (or others like PE, DA, etc), are there any outstanding differences between those reported simulation programs and the one described in this study?
Author Response
- Page 2, line 53: Cited reference-3 is about pneumothorax in ICU. Any citation about studies on (tension) pneumothorax intra-operatively? – in the literature, mortality rates for PTX tend to be in a combination of trauma, ED, and/or ICU settings. We were not able to find any literature on tension PTX solely in an intraoperative setting. We clarified the ICU setting in the manuscript and added another cohort study in this paragraph.
- Page 6: Table 1: Any quantitative analysis and statistical conclusion of the observed results (n=25)? – We did not perform a statistical analysis of the results as the survey was an assessment of the learners’ subjective perceptions of the simulation’s efficacy, rather than an objective assessment. We chose not to grade/stratify residents based on their performance in this simulation, so we do not have specific objective measures to report. Furthermore, a t-test comparing items 1 & 2 to items 3 & 4 would not be appropriate as items 3 & 4 are not assessing a raw confidence level post-simulation, but are rather assessing the degree of improvement as a result of the simulation. Also, with the small n of 25, there is not likely to be a meaningful effect size calculation since the standard of n = 30 is minimum for educational research. We hope to run this again next year and increased the n to 50.
- Page 7, line 292: Any further analysis of the qualitative data? – Only two open-ended responses were recorded and stated verbatim. We plan to ask in-person for qualitative comments next year.
- Page 7, line 311: Indeed, the validity of this simulation protocol remains to be determined. How will the authors validate this teaching model in the future? The authors do indicate two possible limitations in this study. – We added this paragraph to the Discussion section: Of note, the post-survey showed most residents agreed that their confidence level and knowledge increased, and all residents agreed that the debriefing was useful. To follow up longitudinally, we plan to survey the residents as to their experiences and the fre-quency of encountering pneumothorax cases in a variety of clinical settings, including perioperative, OR, ICU, and emergency room. This may serve as validity evidence to support continued simulation sessions, inform realism improvements, and promote self-reflection.
- Are there any similar simulation protocol regarding pre-op and peri-op occurrence of tension pneumothorax for residents in literature? For example,
- American Journal of Roentgenology. 2012;199: 244-251. 10.2214/AJR.11.7892;
- https://doi.org/10.15766/mep_2374-8265.11266
- Adv Simul 1, 21 (2016). https://doi.org/10.1186/s41077-016-0021-2
- MEDICAL EDUCATION 2003;37(Suppl. 1):14–21
- etc
Thank you for the reference list. We did not locate any sources for perioperative occurrence of pneumothorax with respect to anesthesiology simulation education. Perhaps this publication will spark interest and replication at other institutions.
- In addition, since there are similar medical simulation programs for residents to learn tension pneumothorax (or others like PE, DA, etc), are there any outstanding differences between those reported simulation programs and the one described in this study? – for both 5 and 6 – I added a paragraph in the Discussion section (as well as a couple of extra sources) discussing this. The idea I tried to convey is that our simulation places greater emphasis on communication dexterity in the preoperative & perioperative settings (from distractors in Scene 1) as well as the diagnosis of intraoperative PTX via synthesis of data from intraoperative monitoring methods (OR in Scene 2). Other similar simulations seem to be geared more towards developing and assessing the specific competency of chest tube placement.
**********************************************************************************
I have attached the revised manuscript. The revision reflects the changes based on the other reviewer's suggestion as well.
***********************************************************************************

Reviewer 2 Report
I had a pleasure to review an interesting simulation scenario for future anaesthesiologists. As a tutor myself, I find it very insightful, detailed and practical. Although, there are a few minor concerns I have for this manuscript:
1. Introduction:
- abbreviations such as OR, EMT should be given in full when first mentioned. For a non-native speaker it might not be obvious at first what they stand for.
2. Materials and methods:
- I would consider adding the ultrasonogram to the equipment available, maybe the resident in training would think about POCUS in addition to auscultation to provide differential diagnosis?
3. Discussion:
- I personally would find it useful to provide some examples of assertive communication with the surgical team, as each of us has different level of interpersonal skills.
Overall, I find it an excellent example of maintaining proper educational level in a controlled environment so the students can perform better and act more confident in real life. This is a very well-designed scenario that can be implemented in other Simulation Centres worldwide.
Author Response
- Introduction:
- abbreviations such as OR, EMT should be given in full when first mentioned. For a non-native speaker it might not be obvious at first what they stand for. – We clarified these abbreviations per AMA Manual of Style.
- Materials and methods:
- I would consider adding the ultrasonogram to the equipment available, maybe the resident in training would think about POCUS in addition to auscultation to provide differential diagnosis? – Thank you for the suggestion. In the Discussion section, we have included a brief description of the E-FAST exam and the importance to diagnosis and education. This is an improvement opportunity for the future.
- Discussion:
- I personally would find it useful to provide some examples of assertive communication with the surgical team, as each of us has different level of interpersonal skills. –We linked the communication need to the surgeon-anesthesiologist relationship. We have described advocacy and assertive strategies, including the TeamSTEPPS Two-Challenge Rule and CUS.
*************************************************************
Also attached the revised manuscript. This revision also includes the change based on the other reviewer's suggestion.
*************************************************************
